# Heterogeneous and short-term effects of a changing climate on farmers' labor allocation: An empirical analysis of China

**Wolin Zheng[1]ʘ, Xiaozhi Chen[2]ʘ\*, Weiqi Xu[3], Zhidong Wu[3]\***

**1** School of Credit Management, Guangdong University of Finance, Guang Zhou, China, **2** South China Institute of Innovative Finance, Guangdong University of Finance, Guang Zhou, China, **3** College of Economics and Management, South China Agricultural University, Guangzhou, China

ʘ These authors contributed equally to this work.
\* chenxzzz95@163.com (XC); 994521705@qq.com (ZW)

**Data Availability Statement:** Chinese Qiancun Survey is a publicly available dataset. When applying for the right to use the data, the data user should provide true personal information to the

## Abstract

There is growing interest in the impact of climate change on agricultural labor supply in China, rigorous empirical evidence for this issue is insufficient. This potentially important channel through which climate change may affect agricultural labor supply has not received attention. Using a panel survey data of 100 administrative villages and 2977 farmers in China, we find that temperature and precipitation do affect farmers' labor allocation, 1˚C increase from the current average temperature will reduce agricultural labor supply by 0.252%, and 1mm increase from the current average rainfall will reduce agricultural labor supply by 0.001%. Climate change also leads to the decline of net agricultural income, which creates distorted incentives for households to over-supply labor to non-agriculture. Moreover, farmers with relatively lower risk tolerance preferred to reduce the current supply of agricultural labor when net agricultural income is projected to decrease under climate change scenarios.

## 1. Introduction

Climate change harms agricultural production [1, 2]. Climate change cause temperature and precipitation patterns to disrupt the schedule of crop plantation [3, 4]. Previous studies reveal that abnormal climate change is correlated with reducing of crop yield [5, 6]. In areas with long-term unfavorable dryness or wetness conditions, the grain production will inevitably be affected [7, 8]. A 1˚C increase from the current average temperature will reduce grain production by 2% [9]. When the rice temperature exceeds 32˚C in the filling stage, the seed setting rate of rice will be greatly reduced, causing serious production loss [10, 11]. Climate change without CO2 fertilization could reduce the wheat quality in developing countries [12]. In India, climate warming has reduced rice production by 15% and wheat production by 22% [13]. In China, rice production will decrease by 6.1% to 18.6% for every 1˚C increase in temperature [14–16]. Zhang et al. [17] found that average rice yield will decrease 8% in China due to the dramatic increase of average annual temperature without altering the sowing date. If

Survey Data Center of Jinan University (hereinafter referred to as the "Data Provider") and submit the completed "Basic Information of Data User Applicant Form" to the "Data Provider" via iesrsdc@163.com, by fax or by mail to the "Data Provider". Or enter the data platform https://sdc-iesr.jnu.edu.cn/sjzx_15987/list.htm of the Social Survey Center of Jinan University, and according to the website instructions, register and login to the data platform management system to obtain the required data.

**Funding:** The work was supported by the Ministry of Education, Humanities and Social Science Foundation research project (No.23YJC790204) to WZ; and the Guangdong Education Science Planning Project (No. 2023GXJK120) to WZ; and the Guangdong Province Philosophy and Social Science Planning Project (No.GD24YGL08) to WZ; and the Guangdong Province Basic and Applied Basic Research Fund Project (No.2023A1515110314) to WZ; and the Guangzhou Basic and Applied Basic Research Special Project (No.2024A04J3286) to WZ; and the Major projects of National Social Science Found of China (21&ZD090) to ZW; and the Guangdong Province Ordinary University Innovative Team Project (2023WCXTD003) to ZW. There was no additional external funding received for this study.

**Competing interests:** The authors have declared that no competing interests exist.

such climate change continues, many people would be at risk of famine. Climate change is becoming one of the main obstacles to global food security [18].

China is facing significant challenges as a result of climate change, including increased risk of rising temperatures, erratic precipitation, hailstorms, landslides, and glacier shrinkage. These impacts have led to an imbalance in social and economic development. Over the past three decades, China's average temperature has increased more than 1.1°C, a change that is occurring at a much faster rate than the global average. These impacts have led to an imbalance in agricultural systems development. Farmer in particular is highly vulnerable and sensitive to these changes, who are in perpetual threat as farming is the major source of income for their family [19]. According to the traditional economic framework, when climate change poses risks to agricultural production, farmers who are rational economic entities, adjust agricultural production by the rural labor migration. The non-agricultural income obtained through labor migration greatly increases household income and reduces farmers' livelihood vulnerabilities in climate change. Labor reallocation from agriculture to non-agriculture sectors has become a key strategy for farmers to respond to climate change [20]. Historical incidences such as the 1959–1961 Chinese famine highlight the need for farmers' labor allocation to offset negative impacts of climate change [21].

There is a large body of empirical studies assessing the relationship between climate change and labor migration. In line with the New Economics of Labor Migration, many case studies in developing countries reported climate change is causing growing numbers of farmer have given up agricultural production activities and to migrate away from increasingly vulnerable rural [22]. A number of studies highlighted that as of 2020, the number of climate migrants has exceeded 40 million [23]. This trend is expected to accelerate in the coming years, with hundreds of millions of farmer potentially displaced between 2050 and 2100 [24]. Massive out-migration of rural labor forcehad led to significant increase in farmland abandonment and brings challenges in agricultural economy. Governments are beginning to incorporate climate migration in their planning, which has started publishing noteworthy reports on climate migrants [25]. Migration and its relationship to climate change adaptation is undeniable. The increasing in potential risk of climate change aggravates the uncertainty of agricultural production. The income gap between the agricultural sector and the non-agricultural sector has widened. Farmers will reduce agricultural labor input and increase non-agricultural labor input. Huang et al. [26] found that a 1°C increase from the current average temperature will reduce an average rural resident's time allocated to farm work by 7.0%, and increase the time allocated to off-farm work by 7.8%. For rural areas, a stable labor supply is crucial for achieving economic development.

It is worth highlighting that the most existing literature has paid major attention to the impact of cliamte change on the permanent immigration in affected areas. Tubi et al. [27] suggested that migration is often seen either as part of successful adaptation to climate change or as symptomatic of a lack of in-place adaptive capacity. Wang et al. [28] conceptualizes farmer labor migration and its relationship to climate change as forced migration, as farmer's poverty and poor kills mean they have almost no adaptability to climate change, making it difficult for them to adapt and successfully stay in the rural. Jessoe et al. [29] suggested that increased temperatures will lead to a 1.2%-3% decrease in local employment and a 1%-2% increase in domestic migration from rural to urban areas. Climate change has not act as catalysts for permanent immigration in china, in contrast with most developing countries. China established a more transparent system of rural land property rights, farmers were allocated certain farmland areas and received land use rights according to egalitarian principles [30]. For farmer living close to subsistence, the losses from failed migration are such that they are highly averse to moving. Labor migration as a risk-diversifying strategy of china farmer, rather than only as a

response to the income differentials or the relative differences in opportunities between rural and city as in the Neo-classical and push-and-pull theory. On the one hand, when farmers focus on crop production and short-term profits, the negative impact of climate change on agriculture will prompt farmers to given up agricultural production activities and migrated for off-farm work, to reduces farmers' livelihood vulnerabilities. On the other hand, when climate change fail to cause damage to traditional intensive production methods agroecosystems, the expectation of long-term agricultural returns will motivate farmers to proactively adjust labor mobility patterns, to ensure maximum agricultural profits. In this sense, labor migration is a short-term response mechanism for farmers to adapt to the impacts ofclimate change in china.

This means that the existing literature does not pay sufficient attention to the trade-offs farmer has to deal with when allocating manpower resource between cropping activities and off-farm work under the condition of the increasing climate change constraint. Few delve into the perspective of rural labor allocation to systematically research the influence of climate change on the probability and intensity of rural labor supply. Consequently, the existing literature does not assess to what extent the rural labor supply has been maintained and to what extent the labor system has evolved under the condition of cliamte change. Agricultural production takes plants, animals, and microorganisms as objects. The periodicity of agricultural production is determined by animals and plants' growth cycle [31]. Farmers could not thoroughly estimate the loss of farming because of climate change.they have to take the impact of climate change on the agriculture incomes in the previous quarter as the reference point of agriculture labor supply decision. As a rational economic farmer, when faced with the same level of benefits and losses, they will be more sensitive to losses. Farmers with different risk preferences have different levels of agricultural labor supply. Farmers with a higher level of perception about the negative consequences of climate change are more likely to to given up agricultural production activities and migrated for off-farm work. Obviously, agriculture incomes in the previous quarter and farmer's risk preference is a key antecedent to farmer' perceived decisions to labor migration.

Climate change does not directly affect farmer's labor migration, but forms a push for labor migration through its impact on the net agricultural income in the previous quarter and risk preference. In a nutshell, the impact of climate change in agricultural income and loss aversion and how they affect farmers' behavior to withdraw from agriculture activity should be investigated. First, we look at how farmers respond to climate change by allocating agricultural labor supply over the long run. Our findings indicate that climate change affects sectoral agriculture labor allocation, yet labor reallocation in response to climate change is more transitory than permanent. Second, we attempt to quantify the economic significance of these labor reallocation effects by understanding how farmers estimate the probability of climate change. We find that climate change affects risk-averse farmers' labor allocation via net agricultural income in the previous quarter.

This paper aims to make the following contributions. First, it examines the effect of climate change on both the size and pattern of labor migration by farmer in rural China, which will enrich the evaluation of climate change short-term effects. Second, it considers the basic role of net agricultural income in the previous quarter for climate change, that is fundamental to labor migration. More specifically, net agricultural income in the previous quarter caused by climate change will be fully considered when examining the effect of climate change on current decisions to labor migration. This research design will contribute to enhancing our understanding of the climate change-agriculture system in rural china and enable us to measure the relationship between climate change and labor migration more accurately. Third, in view of the decisive role of risk attitude between climate change and labor emigration, a grouping model is constructed. Withthis model, we regard the risk attitude as the grouping variable to

 

explore the indirect effect of cliamte change and net agricultural income in the previous quarter on labor migration. This methodology will clarify the relationships between cliamte change, net agricultural income and labor migration, and thus help us to better understand the dynamics of labor migration under the climate change and agriculture system.

The rest of this paper is organized as follows: Section 2 presents a theoretical analysis of the major channels explaining how climate change affects agricultural labor supply. Section 3 discusses the data, while Section 4 presents the empirical results and robustness checks, and Section 5 discusses the implications of the findings and concludes.

## 2. Literature review and theoretical analysis

### 2.1. Literature review

Global warming is expected to have a significant negative effect on the agricultural sector. This negative effect is likely to increase in the long-term [32]. Therefore, in agriculture, adaptation is an important step [33, 34]. Risk preference plays a central role and is a prerequisite for choosing an adaptation strategy [35]. That requires farmers to have preliminary knowledge about climate variations and their outcomes. If farmers are not fully aware of the climate change risks or the threat of climate change was not serious enough, they would not make responses. Lasco et al. [36] argued that farmers in Philippine who have experience floods had greater climate change risk preference. They adapt to climate change by crop diversification. Elum et al. [37] found that close contact with risk makes farmers focused more on climate change. Because of an enhanced realization of the risks, farmers change in sowing dates or introducing heat-tolerant as a strategy against climate change. Besides changes in planting dates, changes in crop varieties; there are other strategies for adaptation to climate risks. These include, among others, soil and water conservation, irrigation, and fertilization [38, 39]. However, Talanow et al. [40] argued that climate change is a mental construct dependent on farmers's cognition. Assessing the probability of climate risk is pivotal for building farmers' beliefs about climate change. Excessive rainfall and drought exacerbated farmers' anxiety about climate change; rather than seeing the excessive rainfall and drought as evidence to adapt, farmers downplayed the severity of climate change by ascribing the excessive rainfall and drought to natural fluctuations. In short, climate change is essentially an uncertain event. The probability of later events has no matter with previous events [41]. To exactly estimate the probability of climate risk, it needs to systematically analyze the meteorological information such as solar radiation, humidity, wind speed, cloud volume with advanced tools. Farmers do not have such ability, and they could only assess the probability of climate risk according to their experience. Farmers always tend to deviate from the objective facts while they get climate information [42]. For example, in regions with higher concentrations of dryland, farmers hardly recognize the nature of climatic changes. Evidence suggests that farmers do not perceive the effect of climate change, or they misperceive it. Therefore, even if adaptation strategy is an important option to abate the effect of climate change, farmers are not likely to invest in climate change adaptation strategies [43]. Anik et al. [44] found that meteorological records show a significant rainfall increase over the past 30 years in Bangladesh, while the farmers reported a decreasing rainfall trend. As rainfall decreases in the farmers' beliefs, farmers' adopting rate of adaptation strategies decreases. Ojo and Baiyegunhi [45] argued that farmers perceive climate change, but the degree of their perception may vary among farmers. A low level of farmers' climate change perception may be manifest as a result of age, farming experience, and so on [46]. Farming experience plays a vital role in the risk preference of farmers [47]. Water stress can improve wheat quality by concentrating proteins in the grain. Some wheat farmers feel satisfied with harvest when lacking rain [48].

Notably, previous literature focuses on micro-level adaptation strategies. However, there is evidence that climate change micro-level adaptation strategies have not achieved the desired effect [49]. In the absence of government subsidies, even when informed on climate change, farmers might not invest in adaptation strategy, such as improved seeds [50]. The barriers to the adaptation of farm households may be overcome by participating in off-farm activities [51]. Thus, a certain way to reduce exposure to climate change is to reduce agricultural labor input. Bellante [52] showed that corn farmers in Mexico are plagued by climate change. Adaptation strategies to reduce exposure to extreme weather events include using irrigation. If farmers could not afford to invest in irrigation, they may choose to leave the land. Reducing agriculture labor input is the extreme version of adaptation attitude to climate change. This approach means climate change was viewed not only as a problem to solve, but as a rival to be defeated [53]. While this approach is always feasible or even meaningful, its proponents expressed a negative adaptive mindset. Large scale of labor mobility towards non-agricultural sectors may result in exacerbating the extent of abandoned farmland and threatening food security. We recognize this challenge and explicitly attempt the mechanism of climate change and agricultural labor supply to enable adaptation strategies prioritization.

## 2.2. Theoretical derivation

Labor is the significant resource available to farmers, and labor is required for farm and non-farm activities in the household. Farmers allocate labor resources towards activities that bring more returns while not hindering other imperative activities. As a result, the labor' supply is characterized by the united decision of the family numbers. To fulfill livelihood goals, all the family has to allocate part of the labor force in the agricultural sector and other part to the non-agricultural sector. Even some parts of the labor force will be taken as part-time jobs. Over time, because of rising temperatures and reduced precipitation, the crop yield will decrease, and the production costs will increase in China. Climate change has dramatically changed farmers' expectations of agricultural production and non-agricultural employment and breaking the balance of labor distribution. A negative shock in agricultural productivity by climate change reduces labor's marginal productivity, thus cutting down on wages and attracting labor away from the agricultural sector. Climate change leads to an immediate increase in non-agricultural labor demand, thus crowding out agricultural employment. Given these, this study proposes:

Hypothesis1: Climate change has a negative effect on farmers' labor input in agriculture.

Can farmers perceive climate change? It is common sense that climate change is a long-term change in atmospheric conditions only detectable with meteorological instruments. Farmers depend on their short-term experience of climate change without using any devices. Thus, appropriate scientific facts of climate change have not always been directly communicated to farmers. According to Kahneman [54], people would select an estimate as a reference point and then adjust towards the final decision or actual value. The reference point is a neutral outcome, such as the status quo, it will divide the outcomes into gains and losses, which creates risk-bearing and bear significantly upon risk-taking. Thus, the most important thing is to determine the exact location of the reference point. Agriculture takes plants, animals, and microorganisms as objects; the growth cycle of animals and plants determines the periodicity of agriculture. At the end of a complete production cycle, farmers can figure the loss and gains and adjust the labor factors. Farmers assume net agricultural income as potential losses or gains by comparing the impact of climate change on the incomes in the pre-agricultural. According to Thaler and Johnson [55], previous loss urges the actor to avoid the risk and

reduce the adventure behavior, unless the results after the adventure can make up for the loss. Farmers perceive an increasingly salient negative impact of climate change on the the pre-agricultural that leads to increased risk aversion. Farmers avoid investment in human capital to protect their endowed pay values from possible threats. The impact of climate caused the decline of net agricultural income in the previous period, which means risky long-term investments of agriculture might fail. Farmers will have stronger incentives to lock in their positive pay values by reducing the agricultural labor supply. Given these, this study proposes:

Hypothesis 2: Climate change affects farmers' labor input in agriculture via net agricultural income.

Evidence from research in decision-making confirms that an improvement in risk attitude reduces risk aversion, making investors willing to tolerate more risk. Risk attitude represents the expectations of investors relative to a norm. Whatever "norm" may be, it is defined by the propensity to tolerate risks. Consequently, investors with higher risk tolerance are more willing to expose their portfolios to riskier assets in the stock market. That is, the maximum risk loss that farmers can accept subjectively is different. There are differences in farmers' cognition and absorption of risk, which leads to the heterogeneity of risk decision-making [56]. Farmers with lower risk tolerance will take risks only when the risk is small or unlikely to materialize. For these farmers, the expectation of future return from labor in agriculture is downwardly adjusted after accounting for worsened climate status. They are unwilling to take risks that might potentially result in losses. The impact of climate caused the decline of net agricultural income in the previous period, which means risky long-term investments in agriculture might fail. Farmers will reduce the labor supply for agricultural production. The higher risk tolerance changes their subjective assessments about the possibility of various outcomes; they are more likely to focus on positive environmental cues and are willing to take risks. Even if the climate's impact caused the decline of net agricultural income in the previous period, they would not react quickly to avoid losses. Farmers with higher risk tolerance are more willing to take climate risks. They always observe and evaluate risk events for a long time, considering multiple factors (such as the element of supply and demand), and then make a risk decision for labor adjustment [57]. Given these, this study proposes:

Hypothesis 3a: Climate change affects risk-averse farmers' labor input in agriculture via net agricultural income.

Hypothesis 3b: Climate change does not affect risk-loving farmers' labor input in agriculture via net agricultural income.

## 3. Data source and variable selection

### 3.1. Data source

Field surveys data: Individual-level labor allocation comes from the Chinese Qiancun Survey that Jinan University collected. The survey was conducted in 2018, its sampling design aims to make the sample data representative at the provincial level of Guangdong Province. Guangdong province is a major agricultural province in China, undertaking the task of providing food security for the area comprising Guangdong, Hong Kong, and Macao. The area is very hilly, and plains, terraces, platforms, mountains in parts. It is vulnerable to climate change and its impacts as a subtropical monsoon climate zone. The severity of climate change affects the farming of farmers. A large number of farmers poured into the cities for work, resulting in affecting the food security of the area comprising Guangdong, Hong Kong, and Macao Bay.

Therefore, Guangdong province was selected as the research area. The study can serve as the analysis for the mechanism of climate change and agricultural labor supply in hilly and mountainous areas throughout southern China. It uses the PPS sampling method in four stages, with the rural resident population as the auxiliary variable, to sample the counties, administrative villages, natural villages, and farmers in Guangdong Province in turn. The first step is to use the PPS sampling method to select 25 administrative units of districts and counties according to the permanent rural population of all districts and counties under the Guangdong Province's jurisdiction. The second step is to use the PPS sampling method to randomly select four administrative villages belonging to rural areas from each selected administrative unit of districts and counties according to the number of the permanent rural population in each administrative village (100 administrative villages in total). The third step is to take samples of natural villages from each selected administrative unit. The fourth step is to allocate the corresponding number of farmers in each selected administrative village. When the number of natural village questionnaires or farmer questionnaires is insufficient, visitors can apply to the designated person in charge for the supplementary natural village sampling (Jinan University, 2018). The survey covers 25 counties in Guangdong province, 100 administrative villages, and 2977 farmers. The final sample used in this paper includes 1392 farm households, as some important information for 1585 farm households is missing. The Chinese Qiancun Survey was granted ethical approval from the Research Ethics Committees of the Social Science Research Center of Jinan University. All subjects signed written informed consent before the interview.

Climate data: We obtained city-level annual average and precipitation data from 2009 to 2016 from the website of the National Oceanic and Atmospheric Administration (NOAA) of Guangdong province. Climate change at period t-1 affects the number of labor input in agriculture at period t. The variables, like the dependent variable, was measured in 2017 (The survey time is 2018). Therefore, meteorological data time is 2009–2016 to match the data periods. There is at least one weather station with archived observations per city in Guangdong. We extracted data for each city from the weather station located within it. For weather stations within Guangdong, the records after 2006 including temperature, precipitation, and extreme precipitation are recorded and preserved for years. The abrupt changes of temperature and rainfall are obvious that have a more profound impact on the agricultural system. Compared with wind, humidity and other climatic factors, temperature change and rainfall change are easier to be observed and perceived by farmers.

### 3.2. Variable selection

Table 1 list the variables used in this study. Labor input in agriculture, measured by the number of labor engaged in operating arable land in 2017, is the dependent variable [58]. As the independent variable, climate change is measured by rainfall deviation and temperature deviation following the studies by Knapp et al. [59] and Williams [60]. Considering the availability of data and referring to other studies, temperature in 2016 minus the temperature mean in the past eight years and then the absolute value is taken. And rainfall in 2016 minus the rainfall mean in the past eight years and then the absolute value is taken [61]. Net agricultural incomes affected by climate change at period t-1 affect the number of farmers allocated to farm at period t. Consequently, net agricultural incomes in 2016 are the mediation variable. We also control for householder, family, and village characteristics following the literature. Characteristics of householder include gender, age, and education. The family characteristics include family population, land transfer, and land desertification. The village characteristics include soil erosion, benefiting-agriculture policies, traffic development, village economy, irrigation system, land salinization, and industrialization.

**Table 1. Descriptive statistics.**

| Variable | Definition | | Mean | S D | Database |
|---|---|---|---|---|---|
| Labor input in agriculture | Number of family labor engaged in farming activities in 2017 | Actual value (person) | 1.877 | 1.117 | Chinese Qiancun Survey |
| temperature deviation | Temperature in 2016 minus the temperature mean in the past eight years and then the absolute value is taken | Actual value (˚C) | 0.961 | 0.406 | NOAA |
| rainfall deviation | Rainfall in 2016 minus the rainfall mean in the past eight years and then the absolute value is taken | Actual value (mm) | 54.802 | 72.960 | |
| Net agricultural income | Net agricultural income in 2016 | Actual value (thousand Yuan) | 3.715 | 18.879 | Chinese Qiancun Survey |
| Risk tolerance | If 30% of the principal is lost in the investment, can you accept this investment? | 0 = low risk tolerance farmer;1 = high risk tolerance farmers | 0.378 | 0.485 | |
| Gender | Gender of household head | 1 = male;2 = female | 1.149 | 0.356 | |
| Education | Education of household head | 1 = Primary school and below;2 = Junior school;3 = High school;4 = College;5 = Bachelor degree or above | 1.589 | 1.135 | |
| Age | Age of household head | actual value (years old) | 58.160 | 12.209 | |
| Family population | Total family population | Actual value (person) | 4.757 | 2.305 | |
| Land Transfer | Rural land transfer | 0 = No;1 = Yes | 0.010 | 0.095 | |
| Land desertification | The degree of land desertification in your village | 1 = Not Serious 2 = Normal 3 = Serious | 1.314 | 0.628 | |
| Soil erosion | The degree of soil erosion in your village | 1 = Not Serious 2 = Normal 3 = Serious | 1.548 | 0.699 | |
| Benefiting-agriculture policies | Is there a agricultural- oriented policies in your village | 0 = No;1 = Yes | 0.819 | 0.385 | |
| Traffic development | The degree of traffic development in your village | 1 = Very underdeveloped; 2 = underdeveloped; 3 = Normal; 4 = developed; 5 = very developed | 3.614 | 1.027 | |
| village economy | Does your village have any plans to develop sideline business | 0 = No;1 = Yes | 0.418 | 0.493 | |
| Irrigation system | Is there a shortage of irrigation water in your village | 1 = Very short; 2 = shortage; 3 = Normal; 4 = enough; 5 = very abundant | 2.527 | 1.270 | |
| Land salinization | Salinization degree of land in the village | 1 = Not serious;2 = Ordinary;3 = Serious | 1.314 | 0.628 | |
| Degree of industrialization | Is there any factory around the village | 0 = No;1 = Yes | 0.411 | 0.492 | |

## 3.3. Spatial description statistics

According to the administrative villages survey, there are about 961 families in each administrative village, with a permanent population of 4285. The administrative village with the smallest permanent population size is less than 1000, and the largest permanent population size is more than 24000. The average floating population from administrative villages is 1157. And the average number of going out as migrant workers is 979. The east part of Guangdong Province has highest proportion of outflow population, and the lowest is the Pearl River Delta (Table 2).

**Table 2. Population outflow and inflow.**

| Factor | Guangdong Province | Pearl River Delta | East Part of Guangdong Province | West Part of Guangdong Province | North Part of Guangdong Province |
|---|---|---|---|---|---|
| Outflow of population | 1157 | 714 | 1650 | 973 | 1368 |
| number of go out as migrant worker | 979 | 655 | 1367 | 778 | 1170 |
| Inflow of population | 395 | 950 | 176 | 141 | 111 |

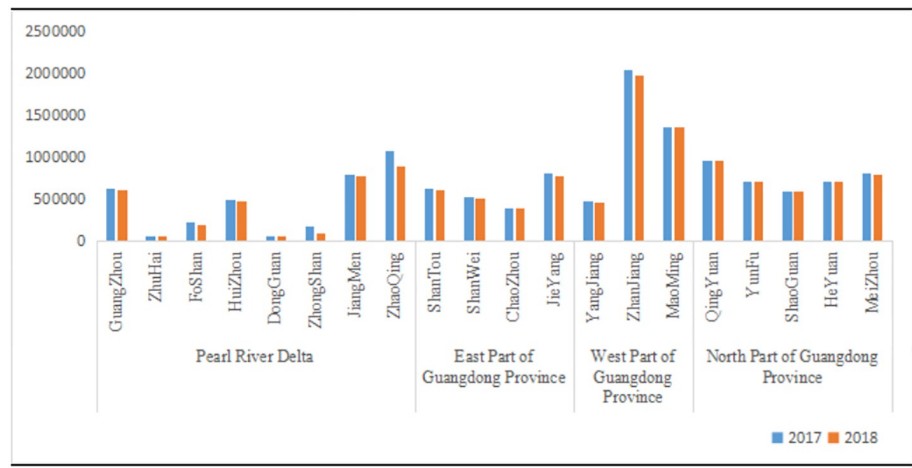

**Fig 1. The agricultural labor force in Guangdong Province.** Note: the data comes from Guangdong Rural Statistical Yearbook.

According to the Guangdong Rural Statistical Yearbook, the highest number of outflow agricultural employment is in Pearl River Delta, and the lowest is in the north part of Guangdong Province. In 2017, there were 3,464,840 labor force in agriculture sectors in the Pearl River Delta. In 2018, there were 3,148,996 labor force in agriculture sectors in the Pearl River Delta. In 2017, there were 2,327,613 labor force in agriculture sectors in the east part of Guangdong Province. In 2018, there were 2,278,320 labor force in agriculture sectors in the east part of Guangdong Province. In 2017, there were 3875618 labor force in agriculture sectors in the west part of Guangdong Province. In 2018, there were 3,787,407 labor force in agriculture sectors in the west part of Guangdong Province. In 2017, there were 3,759,931 labor force in agriculture sectors in the north part of Guangdong Province. In 2018, there were 3,740,321 labor force in agriculture sectors in the north part of Guangdong Province. But except for the cities of Chaozhou, Maoming and Qingyuan, the number of workers employed in agriculture sectors has fallen in the city of Guangdong Province (Fig 1).

## 3.4. Model specification

Our empirical models are as follows:

$$Labor_i = \delta_0 + \delta_1 Climate_i + \delta_2 Control_i + \varepsilon_1 \tag{1}$$

$$Fincome_i = \beta_0 + \beta_1 Climate_i + \beta_2 Control_i + \varepsilon_2 \tag{2}$$

$$Labor_i = \gamma_0 + \gamma_1 Climate_i + \gamma_2 Fincome_i + \gamma_3 Control_i + \varepsilon_3 \tag{3}$$

Where $i$ is the $i^{th}$ farmer; *Control* is the control matrix; $\delta_0$, $\delta_1$, $\delta_2$, $\beta_0$, $\beta_1$, $\beta_2$, $\gamma_0$, $\gamma_2$, $\gamma_3$ are the parameters to be estimated; $\varepsilon_1$, $\varepsilon_2$, $\varepsilon_3$ are the error terms. In addition, Where Eq (1) examines the link between climate change and agricultural labor supply; Eq (2) investigates the relationship between climate change and net agricultural incomes. If $\delta_1$ and $\beta_1$ are significant, Eq (3) continues to be estimated to test whether climate change and net agricultural incomes both related to agricultural labor supply. If $\beta_1$ and $\gamma_2$ are both significant, then there is a mediating effect of net agricultural incomes on the relationship between climate change and agricultural

labor supply. Besides, when $\gamma_1$ is significant, then there is a partial mediating effect. When $\gamma_1$ is not significant, then there is a full mediating effect.

## 4. Results

### 4.1. Climate change and farmer labor allocation

The coefficients on temperature deviation and rainfall deviation are both significantly negative and significant at the 1% level in model 1 of Table 3. Among them, 1˚C increase from the current average temperature will reduce agricultural labor supply by 0.252%, and 1mm increase from current average rainfall will reduce agricultural labor supply by 0.001%. Climate change is challenging to prevent and can inflict considerable damage to arable crops, fruit, flower bulbs, and vegetable. Therefore, climate change can be catastrophic for the farmers. Damage to plants is not reimbursed on the basis of their replacement value. This explains why many farmers decide to reduce agricultural labor supply against recurrent climate variability (e.g., excessive rainfall and drought). Given this, hypothesis1 is verified. Moreover, the coefficient for the family population is positive and significant at the 1% level in model 1 of Table 3, indicating that the family's labor force is given and will not decrease. In the absence of other employment opportunities, farmers tend to invest more labor in agricultural production to digest the surplus workforce of the family. The coefficients on land salinization and industrialization degree are both negative and significant at the 1% level in Table 3 model 1. As expected, the soil salinization of cultivated land is one of the main reasons that caused the low-yielding agriculture.

**Table 3. Climate change and farmers' labor allocation.**

| Variable | Model 1 | | Model 2 | |
|---|---|---|---|---|
| | β-coefficient | Standard Error | β-coefficient | Standard Error |
| temperature deviation | -0.297*** | 0.076 | -0.063 | 0.103 |
| rainfall deviation | -0.002*** | 0.001 | -0.005*** | 0.001 |
| temperature deviation×rainfall deviation | - | - | 0.004*** | 0.001 |
| Gender | 0.194** | 0.082 | 0.210** | 0.083 |
| Education | -0.035 | 0.042 | -0.033 | 0.042 |
| Age | 0.002 | 0.003 | -0.002 | 0.002 |
| Family population | 0.129*** | 0.148 | 0.125*** | 0.015 |
| Land Transfer | -0.631*** | 0.146 | -0.647*** | 0.143 |
| Land desertification | -0.002 | 0.062 | -0.010 | 0.062 |
| Soil erosion | -0.003 | 0.046 | -0.010 | 0.046 |
| Benefiting-agriculture policies | -0.113 | 0.093 | -0.087 | 0.092 |
| Traffic development | -0.019 | 0.033 | 0.001 | 0.034 |
| village economy | -0.014 | 0.057 | -0.003 | 0.059 |
| Irrigation system | -0.003 | 0.023 | -0.013 | 0.023 |
| Land salinization | -0.153*** | 0.062 | -0.146** | 0.062 |
| Degree of industrialization | -0.168*** | 0.067 | -0.212*** | 0.070 |
| Constant | 1.429*** | 0.284 | 0.935*** | 0.291 |
| P-values | 0.000 | | 0.000 | |
| R-squared | 0.124 | | 0.131 | |
| Sample size | 1392 | | 1392 | |

Note:

*, **, *** represent significance levels of 10%, 5% and 1% respectively; standard error in parentheses. And In addition to the survey data comes from rural Chinese households on rural land and related factors of the markets in Table 4, the survey data of Chinese Qiancun Survey are used in Table 3, Tables 5–7.

Farmers tend to invest less labor in agricultural production to cope with land salinization. The process of industrialization allows farmers to seek higher wages. To maximize profits, the young members of labor in farm households have transferred to non-agricultural sectors. Education, age, land desertification, soil erosion, benefiting-agriculture policies, traffic development, village economy, irrigation system of farmers are not statistically significant in explaining the variations in agricultural labor supply. This implies that these variables have no significant effects on farm households' agricultural labor supply.

The interaction coefficient of temperature deviation and rainfall deviation is significant in model 2 of Table 3. As expected, temperature change and rainfall change are distinct. Temperature expresses the degree of cold and heat of the atmosphere. To some extent, it can be expressed as the concept of water. For example, continuous high temperature leads to the decrease of atmospheric water vapor content. Rainfall expresses the concept of water. High temperature may lead to crop failure, but increased rainfall may alleviate this concern. rainfall deviation may reduce production risk cause by temperature deviation, and agricultural labor supply increased relatively.

Robustness check: Some unobserved factors influencing climate change may influence the labor allocation decision of farmers. Endogenous problems may exist between climate change and agricultural labor supply. We employ an instrumental variable (IV) method to reduce the possibility of endogenous problems. This study uses data from rural Chinese households on rural land and related factors of the markets. The dataset was based on a nationally representative household survey conducted from December 2014 to March 2015. We obtain comprehensive information about farm households. Our interviewers used a face-to-face interview method to visit 2880 households. Finally, 2704 questionnaires satisfied our criteria and were used in our study. Farm households' non-agriculture labor input is measured by the proportion of labor engaged in non-agriculture. temperature deviation, the main independent variable in this paper, is measured following the literature. Our IV is the latitude of the county. The higher the latitude, the greater the annual variation of the solar noonday height, the greater the annual variation of the length of day and night; This increases the annual variability in temperature. Therefore, our IV is correlated with our key independent variable. However, latitude has nothing to do with farmers' labor allocation. IV fits the exclusion restriction. We employ a Two-Stage Least Squares (2SLS) regression analysis and reported the estimates in Table 4. The coefficients on latitude are both significantly positive at the 1% level in model 1 of Table 4. Accordingly, temperature deviation increases as the total magnitude of latitude in the county increases. After considering the possible endogenous problems, the results show temperature deviation increases labor demand in non-agriculture.

## 4.2. Climate change, net agricultural income, and farmers' labor allocation

Table 5 presents the results regarding the mediating effect of net agricultural incomes. We first tested the link between climate change and agricultural labor supply, and the results are shown in Step 1 of Table 5. Then we explored the relationship between climate change and net agricultural income, and the results are presented in Step 2 of Table 5. The estimates indicate that climate variables have significantly negative effects on the net agricultural income. Step 3 in Table 5 shows the results regarding the mediating effect of net agricultural incomes. This finding indicates that climate change affects farmers' labor allocation via net agricultural income with all other conditions unchanged. Because of real income declines due to climate change, farmers had a relatively low-value expectation for farming; after comparing this expected value with the possible net agricultural income, they preferred to reduce the agricultural labor supply. In other words, the average net agriculture income of those who adopted the strategies of

 

**Table 4. Results of the robustness.**

| Variable | Model 1 | | Model 2 | |
|---|---|---|---|---|
| | β-coefficient | Standard Error | β-coefficient | Standard Error |
| temperature deviation | - | - | 0.360*** | 0.065 |
| latitude | 0.019*** | 0.001 | - | - |
| Gender | -0.043 | 0.115 | -0.006 | 0.015 |
| Age | 0.001 | 0.001 | -0.002*** | 0.001 |
| Education | 0.018 | 0.109 | 0.031*** | 0.009 |
| Savings | 0.056 | 0.217 | 0.068*** | 0.016 |
| Debt | 0.025 | 0.015 | 0.016 | 0.014 |
| Farmland area | -0.001 | 0.200 | -0.002*** | 0.001 |
| The cultivated land blocks | -0.002 | 0.002 | -0.004*** | 0.002 |
| Non-agricultural technical training | 0.012 | 0.016 | 0.077*** | 0.015 |
| Agricultural technical training | 0.010 | 0.017 | -0.052*** | 0.016 |
| village economy | 0.013 | 0.009 | 0.002 | 0.009 |
| Village terrain | -0.027 | 0.029 | 0.014 | 0.009 |
| Village traffic conditions | 0.009 | 0.008 | 0.038*** | 0.008 |
| Constant | 0.599*** | 0.071 | 0.947*** | 0.096 |
| P-values | 0.000 | | 0.000 | |
| Wald F Atatistic | 0.089 | | 222.572 | |
| DWH Test | - | | 41.117*** | |
| R-squared | - | | 0.103 | |
| Sample size | 2704 | | 2704 | |

Note:

*, **, *** represent significance levels of 10%, 5% and 1% respectively; standard error in parentheses. And the survey data comes from rural Chinese households on rural land and related factors of the markets.

reducing agriculture labor input was significantly lower than that of those who did not. Hypothesis 2 is verified.

## 4.3. Climate change, net agricultural income, and farmers' labor allocation: The heterogeneity of risk tolerance

Table 6 presents the results regarding the mediating effect of net agricultural incomes in the risk-averse farmers. We first tested the link between climate change and agricultural labor supply in the risk-averse farmers, and the results are shown in Step 1 of Table 6. Then we tested the relationship between climate change and net agricultural incomes in the risk-averse farmers, and the results are displayed in Step 2 of Table 5. Step 3 in Table 6 shows the results regarding the mediating effect of net agricultural incomes in the risk-averse farmers. This finding indicates that with all other conditions unchanged, climate change affects labor allocation of lower risk farmers via net agricultural income. Hypothesis 3a is verified.

Table 7 presents the results regarding the mediating effect of net agricultural incomes in the risk-loving farmers. We first tested the relationship between climate change and agricultural labor supply in the risk-loving farmers, and the results are shown in Step 1 of Table 7, where the coefficient of climate change is positive but not significant. Then we investigated the relationship between climate change and net agricultural incomes in the risk-loving farmers, and the results are displayed in Step 2 in Table 7. This finding indicates that coefficient of climate

**Table 5. Climate change, net agricultural income and farmers' labor allocation.**

| Variable | Step 1 | Step 2 | Step 3 |
|---|---|---|---|
| | Agricultural labor supply in Current Period | Net agricultural income in Previous period | Agricultural labor supply in Current Period |
| temperature deviation | -0.297*** (0.076) | -3.714*** (1.284) | -0.325*** (0.080) |
| rainfall deviation | -0.001*** (0.000) | -0.014** (0.007) | -0.001*** (4.108E-04) |
| Net agricultural income in Previous period | - | - | 0.003** (0.002) |
| Gender | 0.194** (0.082) | -4.490 (3.173) | 0.151* (0.091) |
| Education | -0.035 (0.042) | -0.083 (0.690) | -0.023 (0.045) |
| Age | 0.002 (0.003) | -0.046 (0.030) | -0.001 (0.003) |
| Family population | 0.129*** (0.148) | -0.089 (0.233) | 0.125*** (0.016) |
| Land Transfer | -0.631*** (0.146) | 6.131 (6.202) | -0.577*** (0.198) |
| Land desertification | -0.002 (0.062) | -0.837 (0.652) | -0.032 (0.067) |
| Soil erosion | -0.003 (0.046) | 0.977 (0.627) | 0.027 (0.049) |
| Benefiting-agriculture policies | -0.113 (0.093) | -1.371 (2.206) | -0.112 (0.099) |
| Traffic development | -0.019 (0.033) | 0.263 (0.642) | 0.003 (0.035) |
| village economy | -0.014 (0.057) | -0.122 (1.044) | -0.051 (0.061) |
| Irrigation system | -0.003 (0.023) | -0.107 (0.280) | -0.001 (0.024) |
| Land salinization | -0.153*** (0.062) | 0.121 (0.776) | 0.169** (0.070) |
| Degree of industrialization | -0.168*** (0.067) | -2.787* (1.555) | 0.218*** (0.073) |
| constant | 1.429*** (0.284) | 9.904** (4.273) | 1.385*** (0.303) |
| P values | 0.000 | 0.000 | 0.000 |
| R-squared | 0.124 | 0.330 | 0.352 |
| Sample size | 1392 | 1204 | 1204 |

Note:

*, **, *** represent significance levels of 10%, 5% and 1% respectively; standard error in parentheses.

change is positive but not significant. Step 3 in Table 7 shows the mediating effect of net agricultural incomes in the risk-loving farmers. This finding indicates that climate change does not affect lower-risk farmers' labor allocation via net agricultural income with all other conditions unchanged. Hypothesis 3b is verified.

## 5. Conclusions and discussions

A growing body of literature suggests that climate change provides an extra push for farmers to leave the land and to work in non-agricultural sectors. This study explores the mechanism of farmers' labor input in agriculture toward climate change focusing on the farmers in Guangdong Province. The results show that (1) Climate change affects farmers' labor input in agriculture, of which temperature deviation and rainfall deviation have a significant effect. 1°C increase from the current average temperature will reduce agricultural labor supply by 0.252%, and both are of high elasticity. rainfall deviation and farmers' labor input in agriculture have a negative effect with low elasticity. Climate change is more extreme indicates the need for farmers to adopt adaptation strategies of reducing labor input in agriculture. The extent of adaptation undertaken is likely to influence the stability of agriculture in the future. Moreover, temperature deviation and rainfall deviation are not the only elements that affects agricultural labor supply. Other factors, such as gender, family population, land transfer, land salinization, affect farmers' labor input in agriculture. (2) Climate change affects farmers' labor allocation

**Table 6. Climate change, net agricultural income and farmers' labor allocation: Based on risk-averse farmers.**

| Variable | Step 1 | Step 2 | Step 3 |
|---|---|---|---|
| | Agricultural labor supply in Current Period | Net agricultural income in Previous period | Agricultural labor supply in Current Period |
| temperature deviation | -0.312*** (0.096) | -0.990** (0.459) | -0.270*** (0.105) |
| rainfall deviation | -0.002*** (0.001) | -0.009*** (0.003) | -0.001*** (0.001) |
| Net agricultural income in Previous period | - | - | 0.026*** (0.009) |
| Gender | 0.249** (0.112) | 0.446 (0.556) | 0.140 (0.121) |
| Education | -0.080 (0.054) | 0.222 (0.246) | -0.091 (0.056) |
| Age | -0.005 (0.004) | 0.003 (0.017) | -0.005 (0.004) |
| Family population | 0.114*** (0.019) | 0.145 (0.097) | 0.100*** (0.020) |
| Land Transfer | -0.348*** (0.130) | -1.120* (0.648) | -0.348** (0.162) |
| Land desertification | -0.078 (0.079) | -0.404 (0.395) | -0.116 (0.083) |
| Soil erosion | 0.012 (0.063) | -0.375 (0.324) | 0.071 (0.068) |
| Benefiting-agriculture policies | -0.166 (0.127) | 0.899** (0.442) | -0.197 (0.133) |
| Traffic development | 0.002 (0.043) | -0.225 (0.253) | 0.021 (0.045) |
| village economy | -0.007 (0.081) | -0.837** (0.367) | -0.064 (0.083) |
| Irrigation system | 0.035 (0.028) | -0.104 (0.158) | 0.043 (0.028) |
| Land salinization | 0.176** (0.087) | 0.263 (0.477) | 0.185** (0.098) |
| Degree of industrialization | 0.110 (0.092) | -0.608 (0.506) | 0.193** (0.103) |
| constant | 1.689*** (0.385) | 2.494 (2.115) | 1.644*** (0.400) |
| P values | 0.000 | 0.000 | 0.000 |
| R-squared | 0.123 | 0.130 | 0.138 |
| Sample size | 837 | 733 | 733 |

Note:

*, **, *** represent significance levels of 10%, 5% and 1% respectively; standard error in parentheses.

via net agricultural income. In other words, net agricultural income is the primary factor in farmers' labor input in agriculture. Thus, by adjusting the net agricultural income, we can improve agricultural labor supply. (3) Risk tolerance affects farmers' perception of climate change. Active responders of climate change are those with lower risk tolerance. If farmers are risk-averse, they will completely prefer a certain outcome to an uncertain one with the same expected value. Farmers with different levels of loss risks have different decisions on agricultural labor supply. Climate change affects risk-averse farmers' labor allocation via net agricultural income. However, farmers with higher risk tolerance are not sensitive to climate change. Climate change does not affect risk-loving farmers' labor allocation via net agricultural income.

Because farmers are unlikely to precisely seize the adaptation strategies to protect themselves against climate change, hence, farmers protect themselves against climate change by reducing agricultural labor supply. In other words, the existing of farmer's adaptation strategies developed in response to climate change such as heat waves, droughts, and floods may not be enough. More transformative climate adaptation may be required. We can put forward the following proposals to improve the effectiveness of adaptation strategies. (1) Risk assessment and information release system may also be valued as a climate risk-mitigation strategy. Indeed, risk assessment plays multiple roles, especially in smallholder farming systems. But it needs to build on robust methods of designing. We should improve the integration and utilization of agricultural emergency management data and take the administrative village as the

**Table 7. Climate change, net agricultural income and farmers' labor allocation: Based on Risk-loving farmers.**

| Variable | Step 1 | Step 2 | Step 3 |
|---|---|---|---|
| | Agricultural labor supply in Current Period | Net agricultural income in Previous period | Agricultural labor supply in Current Period |
| temperature deviation | -0.197 (0.132) | -5.194** (2.113) | -0.319** (0.129) |
| rainfall deviation | -0.001 (0.001) | -0.022 (0.018) | 0.000 (0.001) |
| Net agricultural income in Previous period | - | - | 0.005*** (0.001) |
| Gender | 0.080 (0.130) | -9.793** (4.684) | 0.124 (0.151) |
| Education | 0.074 (0.071) | -2.728 (2.209) | 0.136* (0.075) |
| Age | 0.003 (0.004) | -0.100 (0.074) | 0.005 (0.004) |
| Family population | 0.148*** (0.026) | -0.356 (0.393) | 0.160*** (0.027) |
| Land Transfer | -1.014*** (0.261) | 17.687 (11.931) | -1.271** (0.531) |
| Land desertification | 0.055 (0.102) | -0.244 (1.563) | 0.059 (0.110) |
| Soil erosion | -0.041 (0.072) | 1.657 (1.228) | -0.035 (0.074) |
| Benefiting-agriculture policies | 0.003 (0.144) | -0.283 (2.285) | -0.006 (0.157) |
| Traffic development | -0.036 (0.055) | 0.490 (1.117) | -0.003 (0.058) |
| village economy | -0.043 (0.097) | 2.566 (2.451) | -0.045 (0.101) |
| Irrigation system | -0.065 (0.043) | 0.017 (0.764) | -0.085* (0.046) |
| Land salinization | 0.139* (0.082) | 1.795 (1.510) | 0.136 (0.090) |
| Degree of industrialization | 0.295*** (0.102) | -6.526** (3.196) | 0.302*** (0.109) |
| constant | 0.973*** (0.417) | 11.702 (7.282) | 0.894** (0.460) |
| P values | 0.000 | 0.000 | 0.000 |
| R-squared | 0.133 | 0.592 | 0.167 |
| Sample size | 506 | 431 | 431 |

Note:

*, **, *** represent significance levels of 10%, 5% and 1% respectively; standard error in parentheses.

unit, establish the risk management organization with the village committee and grassroots cadres, and formulate the emergency management plan with the actual situation. The emergency management plan will need to account for the potential impacts of climate change, including those with low probability but significant consequences. (2) Governments should improve fiscal transfer income for the agricultural insurance and increase rural residents' property income to reduce farmers' pressure on agricultural insurance. Agricultural insurance subsidies only provide to farmers who promote sustainable agriculture, including investment in efficient fertilizer application. The government should establish teaching resource sharing to enhance agricultural insurance recognition of farmers. And agricultural insurance contracts and trigger levels that should be consciously designed may be offered on the village rather than individual farmers.

Notably, China is one of the countries in the world with a large distribution area of ecologically fragile areas and complex types of ecologically fragile surfaces. Climate change will exacerbate ecological stress for the farmers to leave the land, especially for the impoverished ones [62]. Agricultural systems were disturbed by labor out-migration, that can not perform an important security function, providing a degree of safe shelter to farmers in china [63]. If farmers are unable to adapt to new ways of switching manpower resource, they may encounter challenges in the off-farm jobs. We highlight the critical need for future research to examine the role of ecological factors in shaping migration responses to climate change, and the impact of labor migration on farmer's welfare under the background of climate change. Moreover, the

large-scale rural-to-urban labor migration has brought forth profound social and economic changes [64]. On the one hand, rural out-migration naturally leads to agricultural land abandonment. And rural households with labor migrants had lower agricultural productivity than those without migratory workers [65]. On the other hand, regional labor distributions are also disrupted by increased climate change. Climate change exacerbation reduces wages, and the welfare of the local labor market. And the magnitude of these impacts are uncertainties and warrant further research. Further research focuses on the impact of farmer migration decisions on food security, agricultural production efficiency, and local labor market distribution in the context of climate change.

It is an open question whether the influence of previous agricultural labor supply experience results from exposure to the climate-related hazard or the actual input decisions in agriculture. This study does not observe whether the farmers who are previously exposed to the climate-related hazard. We do not separate those who have not reduced labor input in agriculture but thought about the threat of climate change from those who have not thought about the threat of climate change. Nevertheless, we suspect that cost of risk-averse may backfire amongst farmers who are pre-defined preferences across alternatives. Farmers in the opt-out condition are likely to be more focused on the potential climate change losses. Farmers in the opt-out condition are more likely to regret reducing labor input in agriculture if a climate-related hazard does not happen. Without detailed information on farmers, such as their preferences towards different climate-related hazards, any attempt to increase farmers' labor input in agriculture may be ineffective. However, required time series data for village levels' net agricultural income were unavailable at the Chinese Qiancun Survey, hence our improvisation for data acquisition. There are limitations in terms of survey data length and sample size, which our findings' generalizability. Further, improvements of the study are achievable using comprehensive time-series data at village levels. Since comprehensive time-series data at village levels are unavailable from survey data, a field experiment could be designed to obtain annual primary data for increased accuracy of information from farmers. It would be very helpful if future research could include climate-related hazard projection (estimation of basis risks) and farmers' preference towards different climate-related hazards.

## Supporting information

**S1 File.**
(RAR)

## Author Contributions

**Conceptualization:** Wolin Zheng, Zhidong Wu.

**Data curation:** Wolin Zheng, Xiaozhi Chen.

**Formal analysis:** Weiqi Xu, Zhidong Wu.

**Methodology:** Wolin Zheng, Xiaozhi Chen.

**Project administration:** Zhidong Wu.

**Supervision:** Weiqi Xu, Zhidong Wu.

**Writing – original draft:** Wolin Zheng, Zhidong Wu.

**Writing – review & editing:** Wolin Zheng, Xiaozhi Chen, Zhidong Wu.

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
