## [Decision Letter · Decision Letter 0]

6 May 2024

PONE-D-24-02544Farmers’ Labor Allocation Response to Climate Change in ChinaPLOS ONE

Dear Dr. Wu,

Thank you for submitting your manuscript to PLOS ONE. After careful consideration, we feel that it has merit but does not fully meet PLOS ONE’s publication criteria as it currently stands. Therefore, we invite you to submit a revised version of the manuscript that addresses the points raised during the review process.

We look forward to receiving your revised manuscript.

Kind regards,

Zulqarnain Mushtaq, PhD

Academic Editor

PLOS ONE

Journal Requirements:

The work was supported by Guangdong Education Science Planning Project(No. 2023GXJK120);Guangdong Federation of Social Science will support provincial social science organizations to carry out research projects in 2023 (No. GD2023SKFC19); and the Major projects of National Social Science Found of China (21&ZD090)； Guangdong Youth Research Project (No. 2022GJ005).

"N/A"

5. Please provide a complete Data Availability Statement in the submission form, ensuring you include all necessary access information or a reason for why you are unable to make your data freely accessible. If your research concerns only data provided within your submission, please write "All data are in the manuscript and/or supporting information files" as your Data Availability Statement.

6. We note that Figures 2 and 3 in your submission contain map images which may be copyrighted. All PLOS content is published under the Creative Commons Attribution License (CC BY 4.0), which means that the manuscript, images, and Supporting Information files will be freely available online, and any third party is permitted to access, download, copy, distribute, and use these materials in any way, even commercially, with proper attribution. For these reasons, we cannot publish previously copyrighted maps or satellite images created using proprietary data, such as Google software (Google Maps, Street View, and Earth). For more information, see our copyright guidelines: http://journals.plos.org/plosone/s/licenses-and-copyright.

We require you to either present written permission from the copyright holder to publish these figures specifically under the CC BY 4.0 license, or remove the figures from your submission:

a. You may seek permission from the original copyright holder of Figures 2 and 3 to publish the content specifically under the CC BY 4.0 license.  

Reviewers' comments:

Reviewer's Responses to Questions

**Comments to the Author**

1. Is the manuscript technically sound, and do the data support the conclusions?

Reviewer #1: Yes

2. Has the statistical analysis been performed appropriately and rigorously? 

Reviewer #1: Yes

3. Have the authors made all data underlying the findings in their manuscript fully available?

Reviewer #1: Yes

4. Is the manuscript presented in an intelligible fashion and written in standard English?

Reviewer #1: Yes

5. Review Comments to the Author

**Reviewer #1**: 1. Firstly, the title needs improvement to better represent the study's purpose.

2. The introduction section appears to be somewhat weak and could benefit from further elaboration. It is essential to highlight the importance of the subject area and provide a clear understanding of the problem. Additionally, including a theoretical framework for the study is recommended. Moreover, I suggest adding more relevant literature and citations to strengthen the introduction.

3. There is a significant need to more elaborate the discussion section in the paper with latest references.

4. The conclusion section could benefit from the addition of information regarding limitations and suggestions for future research.

5. In general, I found the manuscript interesting, but it seems somewhat simplistic in its current form. Authors should incorporate statistical techniques and include more citations to enhance the paper's quality.

6. PLOS authors have the option to publish the peer review history of their article (what does this mean?). If published, this will include your full peer review and any attached files.

Reviewer #1: No

---

## [Author Response · Author response to Decision Letter 0]

27 May 2024

Detailed Response to Reviewers

Dear Editors, Dr. Zulqarnain Mushtaq and Reviewers:

Thank you for your letter and for the reviewers’ comments concerning our manuscript entitled "Farmers’ Labor Allocation Response to Climate Change in China" (ID: PONE-D-24-02544). Those comments are all valuable and very helpful for revising and improving our paper, as well as the important guiding significance to our researches. We have studied comments carefully and have made correction which we hope meet with approval. We upload two manuscript, one of which is a separate file labeled 'Revised Manuscript with Track Changes' and the other one is a separate file labeled 'Manuscript'. Revised portion are marked in red in the paper. The main corrections in the paper and the responds to the comments are as following: 

1. When submitting your revision, we need you to address these additional requirements.Please ensure that your manuscript meets PLOS ONE's style requirements, including those for file naming. 

We carefully read the PLOS ONE style templates and edit our manuscript according to the guidelines for main body, affiliations and file naming to ensure our revised manuscript meets PLOS ONE's style requirements.

2. We note that the grant information you provided in the ‘Funding Information’ and ‘Financial Disclosure’ sections do not match.When you resubmit, please ensure that you provide the correct grant numbers for the awards you received for your study in the ‘Funding Information’ section.

3. Thank you for stating in your Funding Statement:Please provide an amended statement that declares all the funding or sources of support (whether external or internal to your organization) received during this study, as detailed online in our guide for authors at http://journals.plos.org/plosone/s/submit-now. Please also include the statement “There was no additional external funding received for this study.” in your updated Funding Statement.Please include your amended Funding Statement within your cover letter. We will change the online submission form on your behalf.

We carefully read a lot of articles from PLOS ONE and modified our funding statement. 

“Funding: The work was supported by the Ministry of Education, Humanities and Social Science Foundation research project (No.23YJC790204); and the Guangdong Education Science Planning Project (No. 2023GXJK120); and the Guangdong Province Philosophy and Social Science Planning Project (No.GD24YGL08); and the Guangdong Province Basic and Applied Basic Research Fund Project (No.2023A1515110314); and the Guangzhou Basic and Applied Basic Research Special Project (No.2024A04J3286); and the Major projects of National Social Science Found of China (21&ZD090); and the Guangdong Province Ordinary University Innovative Team Project (2023WCXTD003). There was no additional external funding received for this study.”

"N/A" 

We carefully read the PLOS ONE's Competing Interests policy and modified our Competing Interests statement.

Competing Interests statement: The authors have declared that no competing interests exist.

5. Please provide a complete Data Availability Statement in the submission form, ensuring you include all necessary access information or a reason for why you are unable to make your data freely accessible. If your research concerns only data provided within your submission, please write "All data are in the manuscript and/or supporting information files" as your Data Availability Statement.

We carefully read a lot of articles from PLOS ONE and modified our data availability statement. 

Data availability：Chinese Qiancun Survey is a publicly available dataset. When applying for the right to use the data, the data user should provide true personal information to the Survey Data Center of Jinan University (hereinafter referred to as the "Data Provider") and submit the completed "Basic Information of Data User Applicant Form" to the "Data Provider" via iesrsdc@163.com , by fax or by mail to the "Data Provider". Or enter the data platform https://sdc-iesr.jnu.edu.cn/sjzx_15987/list.htm of the Social Survey Center of Jinan University, and according to the website instructions, register and login to the data platform management system to obtain the required data.

6.We note that Figures 2 and 3 in your submission contain map images which may be copyrighted. 

We remove the figure 2 and 3 in our new submission.

We carefully checked and modified the reference list to ensure that it is complete and correct.

Response to Reviewer #1:

Thank you for your careful and thoughtful examination of our paper. Incorporating your comments, as well as those of the Editor, and the other referee, resulted in a greatly improved manuscript. We address each of your comments below.

1. Firstly, the title needs improvement to better represent the study’s purpose. 

We thank the referee for this helpful comment. We changed the original title to read: Heterogeneous and short-term effects of a changing climate on farmers’ labor allocation: An empirical analysis of China.

2. The introduction section appears to be somewhat weak and could benefit from further elaboration. It is essential to highlight the importance of the subject area and provide a clear understanding of the problem. Additionally, including a theoretical framework for the study is recommended. Moreover, I suggest adding more relevant literature and citations to strengthen the introduction.

We thank the referee for this helpful comment. The revised introduction is as follows.

Climate change harms agricultural production[1,2]. Climate change cause temperature and precipitation patterns to disrupt the schedule of crop plantation[3,4]. Previous studies reveal that abnormal climate change is correlated with reducing of crop yield[5,6]. In areas with long-term unfavorable dryness or wetness conditions, the grain production will inevitably be affected [7,8]. A 1°C increase from the current average temperature will reduce grain production by 2%[9]. When the rice temperature exceeds 32℃ in the filling stage, the seed setting rate of rice will be greatly reduced, causing serious production loss[10,11]. Climate change without CO2 fertilization could reduce the wheat quality in developing countries[12]. In India, climate warming has reduced rice production by 15% and wheat production by 22%[13]. In China, rice production will decrease by 6.1% to 18.6% for every 1℃ increase in temperature [14-16]. Zhang et al. [17]found that average rice yield will decrease 8% in China due to the dramatic increase of average annual temperature without altering the sowing date. If such climate change continues, many people would be at risk of famine. Climate change is becoming one of the main obstacles to global food security[18].

China is facing significant challenges as a result of climate change, including increased risk of rising temperatures, erratic precipitation, hailstorms, landslides, and glacier shrinkage. These impacts have led to an imbalance in social and economic development. Over the past three decades, China’s average temperature has increased more than 1.1℃, a change that is occurring at a much faster rate than the global average. These impacts have led to an imbalance in agricultural systems development.Farmer in particular is highly vulnerable and sensitive to these changes, who are in perpetual threat as farming is the major source of income for their family[19].According to the traditional economic framework, when climate change poses risks to agricultural production, farmers who are rational economic entities, adjust agricultural production by the rural labor migration.The non-agricultural income obtained through labor migration greatly increases household income and reduces farmers' livelihood vulnerabilities in climate change. Labor reallocation from agriculture to non-agriculture sectors has become a key strategy for farmers to respond to climate change[20].Historical incidences such as the 1959-1961 Chinese famine highlight the need for farmers’ labor allocation to offset negative impacts of climate change[21].

There is a large body of empirical studies assessing the relationship between climate change and labor migration.In line with the New Economics of Labor Migration, many case studies in developing countries reported climate change is causing growing numbers of farmer have given up agricultural production activities and to migrate away from increasingly vulnerable rural[22]. A number of studies highlighted that as of 2020, the number of climate migrants has exceeded 40 million[23].This trend is expected to accelerate in the coming years, with hundreds of millions of farmer potentially displaced between 2050 and 2100[24]. Massive out-migration of rural labor forcehad led to significant increase in farmland abandonment and brings challenges in agricultural economy.Governments are beginning to incorporate climate migration in their planning, which has started publishing noteworthy reports on climate migrants[25]. Migration and its relationship to climate change adaptation is undeniable. The increasing in potential risk of climate change aggravates the uncertainty of agricultural production.The income gap between the agricultural sector and the non-agricultural sector has widened.Farmers will reduce agricultural labor input and increase non-agricultural labor input. Huang et al.[26] found that a 1℃ increase from the current average temperature will reduce an average rural resident's time allocated to farm work by 7.0%, and increase the time allocated to off-farm work by 7.8%. For rural areas, a stable labor supply is crucial for achieving economic development. 

It is worth highlighting that the most existing literature has paid major attention to the impact of cliamte change on the permanent immigration in affected areas. Tubi et al.[27] suggested that migration is often seen either as part of successful adaptation to climate change or as symptomatic of a lack of in-place adaptive capacity.Wang et al.[28] conceptualizes farmer labor migration and its relationship to climate change as forced migration, as farmer's poverty and poor kills mean they have almost no adaptability to climate change, making it difficult for them to adapt and successfully stay in the rural. Jessoe et al.[29] suggested that increased temperatures will lead to a 1.2%-3% decrease in local employment and a 1%-2% increase in domestic migration from rural to urban areas.Climate change has not act as catalysts for permanent immigration in china, in contrast with most developing countries. China established a more transparent system of rural land property rights,farmers were allocated certain farmland areas and received land use rights according to egalitarian principles[30].For farmer living close to subsistence, the losses from failed migration are such that they are highly averse to moving. Labor migration as a risk-diversifying strategy of china farmer, rather than only as a response to the income differentials or the relative differences in opportunities between rural and city as in the Neo-classical and push-and-pull theory. On the one hand,when farmers focus on crop production and short-term profits, the negative impact of climate change on agriculture will prompt farmers to given up agricultural production activities and migrated for off-farm work, to reduces farmers' livelihood vulnerabilities. On the other hand, when climate change fail to cause damage to traditional intensive production methods agroecosystems, the expectation of long-term agricultural returns will motivate farmers to proactively adjust labor mobility patterns, to ensure maximum agricultural profits. In this sense, labor migration is a short-term response mechanism for farmers to adapt to the impacts of climate change in china.

This means that the existing literature does not pay sufficient attention to the trade-offs farmer has to deal with when allocating manpower resource between cropping activities and off-farm work under the condition of the increasing climate change constraint. Few delve into the perspective of rural labor allocation to systematically research the influence of climate change on the probability and intensity of rural labor supply.Consequently, the existing literature does not assess to what extent the rural labor supply has been maintained and to what extent the labor system has evolved under the condition of cliamte change.Agricultural production takes plants, animals, and microorganisms as objects.The periodicity of agricultural production is determined by animals and plants' growth cycle[31]. Farmers could not thoroughly estimate the loss of farming because of climate change.they have to take the impact of climate change on the agriculture incomes in the previous quarter as the reference point of agriculture labor supply decision.As a rational economic farmer, when faced with the same level of benefits and losses, they will be more sensitive to losses. Farmers with different risk preferences have different levels of agricultural labor supply. Farmers with a higher level of perception about the negative consequences of climate change are more likely to to given up agricultural production activities and migrated for off-farm work. Obviously, agriculture incomes in the previous quarter and farmer’s risk preference is a key antecedent to farmer’ perceived decisions to labor migration.

Climate change does not directly affect farmer’s labor migration, but forms a push for labor migration through its impact on the net agricultural income in the previous quarter and risk preference.In a nutshell, the impact of climate change in agricultural income and loss aversion and how they affect farmers’ behavior to withdraw from agriculture activity should be investigated.First, we look at how farmers respond to climate change by allocating agricultural labor supply over the long run. Our findings indicate that climate change affects sectoral agriculture labor allocation, yet labor reallocation in response to climate change is more transitory than permanent.Second, we attempt to quantify the economic significance of these labor reallocation effects by understanding how farmers estimate the probability of climate change.We find that climate change affects risk-averse farmers’ labor allocation via net agricultural income in the previous quarter.

This paper aims to make the following contributions.First, it examines the effect of climate change on both the size and pattern of labor migration by farmer in rural China, which will enrich the evaluation of climate change short-term effects. Second, it considers the basic role of net agricultural income in the previous quarter for climate change,that is fundamental to labor migration. More specifically, net agricultural income in the previous quarter caused by climate change will be fully considered when examining the effect of climate change on current decisions to labor migration.This research design will contribute to enhancing our understanding of the climate change-agriculture system in rural china and enable us to measure the relationship between climate change and labor migration more accurately. Third, i

---

## [Editor Report · Decision Letter 1]

14 Jun 2024

Heterogeneous and short-term effects of a changing climate on farmers’ labor allocation: An empirical analysis of China

PONE-D-24-02544R1

Dear Dr. Wu,

We’re pleased to inform you that your manuscript has been judged scientifically suitable for publication and will be formally accepted for publication once it meets all outstanding technical requirements.

Kind regards,

Zulqarnain Mushtaq, PhD

Academic Editor

PLOS ONE
---

## [Editor Report · Acceptance letter]

18 Jul 2024

PONE-D-24-02544R1 

PLOS ONE

Dear Dr. Wu, 

I'm pleased to inform you that your manuscript has been deemed suitable for publication in PLOS ONE. Congratulations! Your manuscript is now being handed over to our production team.

Kind regards, 

on behalf of

Dr. Zulqarnain Mushtaq 

Academic Editor

PLOS ONE